# The Longitudinal Impact of Father Presence on Adolescent Depressive Symptoms: The Mediating Role of Emotion Beliefs and Emotion Regulation

**DOI:** 10.3390/bs16010047

**Published:** 2025-12-25

**Authors:** Dan Xu, Haowen Peng, Zongkui Zhou, Jing Wang

**Affiliations:** 1School of Psychology, Central China Normal University, Wuhan 430079, China; dannel@mails.ccnu.edu.cn (D.X.); penghaowen0223@mails.ccnu.edu.cn (H.P.); zhouzk@ccnu.edu.cn (Z.Z.); 2Key Laboratory of Adolescent Cyberpsychology and Behavior (CCNU), Ministry of Education, Wuhan 430079, China; 3Key Laboratory of Human Development and Mental Health of Hubei Province, Wuhan 430079, China

**Keywords:** father presence, emotion beliefs, emotion regulation, adolescent depressive symptoms

## Abstract

Background: Adolescence is a developmental period marked by heightened vulnerability to depressive symptoms. Although prior research highlights the significance of father presence in adolescent mental health, longitudinal evidence clarifying both its direct and indirect effects remains scarce. Methods: The present study used a three-wave longitudinal design to examine whether emotion beliefs and emotion regulation processes explain the link between father presence and depressive symptoms. Participants included 1074 Chinese adolescents (*M_age_* = 16.06, *SD* = 0.43, girls = 52.89%). Results: Path models showed that higher perceived father presence predicted lower depressive symptoms over time. Emotion beliefs and cognitive reappraisal each served as significant mediators in this association. Moreover, a sequential pathway emerged that father presence predicted fewer maladaptive emotion beliefs, which in turn were associated with the use of cognitive reappraisal, ultimately reducing depressive symptoms. Conclusions: These findings shed light on the cognitive and regulatory processes through which paternal presence contributes to adolescent emotional adjustment and provide support for incorporating paternal emotional engagement and emotion socialization strategies into family-based prevention and intervention programs targeting adolescent depression.

## 1. Introduction

Adolescence is a pivotal developmental stage characterized by rapid biological, psychological, and social role transitions, during which vulnerability to depressive symptoms markedly increases ([32]; [50]). Globally, more than one-fifth of children and adolescents experience a depressive episode or display depressive symptoms ([44]), and the prevalence of these symptoms continues to rise during adolescence ([43]). Depression during adolescence is associated with impairments in academic functioning, interpersonal relationships, and social adjustment, and also increases the risk of self-injury and suicide, making it a pressing public health concern ([41]). From an ecological systems perspective, the family serves the core microsystem in which children’s development unfolds. Parents’ behaviors exert direct and enduring influences on children’s psychological and social adjustment through daily interactions ([4]). Empirical evidence consistently indicates a strong association between family functioning and adolescents’ depressive symptoms ([30]), perhaps because adolescents raised in warm and supportive family environments are more likely to develop adaptive emotion regulation strategies and greater psychological resilience ([58]).

In many cultural contexts, including China, caregiving responsibilities largely fall on mothers, while father absence or limited paternal involvement is relatively common ([39]). Such patterns may pose potential risks to adolescents’ mental health ([15]; [28]). Although previous research has shown a significant negative relationship between father presence, defined as adolescents’ perceptions of their fathers’ emotional investment and psychological availability, and depressive symptoms among adolescents ([48]; [49]), studies examining both the direct and indirect effects of father presence on adolescent depressive symptoms remain limited. Moreover, the underlying mediating mechanisms through which father presence influences adolescent depressive symptoms have yet to be elucidated.

Within this framework, emotion beliefs and emotion regulation may serve as key mediating pathways linking father presence to adolescents’ mental health. Prior studies provide initial support showing that fathers’ positive emotional involvement and socio-emotional behaviors shape youths’ emotion beliefs and influence their use of adaptive emotion regulation strategies ([21]; [60]). Yet, longitudinal research testing these mechanisms remains scarce. To address these gaps, the present study employs a longitudinal design to examine how father presence relates to later depressive symptoms in adolescence, and to test whether this association is explained by adolescents’ emotion beliefs and their use of emotion regulation strategies. Guided by theory, we test two types of mediating pathways, including independent indirect effects via emotion beliefs and via emotion regulation, and a theoretically ordered sequence in which adaptive emotion beliefs relate to better emotion regulation strategies. By clarifying these mechanisms, this study seeks to articulate how perceptions of father presence are translated into emotional competencies that may protect adolescents from depressive symptoms, providing a more precise theoretical framework to inform father-focused prevention and intervention efforts.

### 1.1. The Relationship Between Father Presence and Depressive Symptoms

The theory of father presence ([38]) offers a multifaceted explanation of children’s perceptions of paternal affection, including father–son relationships, intergenerational family dynamics, and children’s belief systems about the role of the father. In this framework, father presence is conceptualized as a psychological experience rather than a physical or behavioral indicator. It refers to the child’s subjective sense that the father is emotionally available, responsive, and perceived engaged. This conceptualization emphasizes the internal psychological experience of being emotionally supported by one’s father, which is particularly relevant for adolescents’ cognitive and emotional processing. Empirical research indicates that higher perceived paternal presence is linked to more positive emotional experiences, stronger self-esteem, and better overall psychological adjustment among children and adolescents ([48]; [49]). [49] ([49]), for instance, conducted a systematic review and found that active father involvement in childrearing during early childhood is conducive to the promotion of children’s social competence, emotional regulation, and secure attachment, thereby establishing the foundations for future psychological well-being. Conversely, extended periods of paternal absence, neglectful parenting, or emotional disengagement have been linked to emotional disturbances in adolescents, behavioral problems, and social adjustment difficulties ([12]; [14]). These patterns may disrupt the emotional support system within the family and weaken a child’s sense of security and efficacy, leaving them more vulnerable to depressive experiences.

Findings across cultural contexts further support the protective role of fathers. For instance, [9] ([9]) reported that even in families where mothers were at high risk of depression, active father involvement remained significantly associated with fewer behavioral problems in children, confirming the importance of the father’s role beyond the influence of the mother. Similarly, [62] ([62]) observed that the presence of the father contributes to the psychological security and emotional stability of children, thereby enhancing their capacity to cope with stress and emotional challenges. Together, these findings imply that father presence contributes not only to family functioning but also to children’s psychological resilience. Based on this body of work, we expected that higher perceived father presence would predict fewer depressive symptoms over time.

### 1.2. The Mediating Role of Emotion Beliefs

Emotion beliefs refer to an individual’s cognitive assessment of emotional flexibility, specifically their perception of the possibility of changing, managing, and controlling emotions ([56]). Emotion beliefs typically comprise two fundamental dimensions, including how controllable and functional emotions are ([3]). Adolescents who view emotions as controllable and meaningful tend to report higher psychological well-being and fewer depressive symptoms ([21]; [55]). Conversely, beliefs that emotions are fixed or inherently unhelpful are associated with reduced motivation to regulate emotions and greater vulnerability to depression ([20]; [33]; [35]; [36]).

Adolescence is a formative period for developing emotion beliefs, and the family environment plays a central role in shaping how youth conceptualize emotions. Through observation, emotional social communication, and direct guidance on expressing and regulating emotions, adolescents develop beliefs about whether emotions can be understood and managed ([42]). Recent work suggests that fathers may be especially influential in the formation of adaptive emotion beliefs. Adolescents who perceive their fathers as emotionally present and available tend to appraise emotions as less threatening and more manageable ([6]; [11]). Indeed, father presence seem to foster positive emotion beliefs, contributing to a more adaptive understanding of emotional experiences ([10]). On this basis, we proposed that emotion beliefs would mediate the relationship between father presence and depressive symptoms in adolescents.

### 1.3. The Mediating Role of Emotion Regulation Strategies

Emotion regulation is the capacity to monitor, evaluate, and modify one’s emotional experiences and expressions across various contexts, representing a crucial internal resource for psychological well-being ([23]). A substantial body of evidence indicates that individual differences in emotion regulation are closely linked to internalizing difficulties, including depressive symptoms. Specifically, maladaptive strategies such as expressive suppression tend to correlate with elevated depressive symptoms, whereas adaptive approaches, like cognitive reappraisal, are associated with lower levels of depressive symptoms ([22]; [30]; [54]).

Parents are widely acknowledged as the main agents of socialization, playing a central role in the development of children’s emotion regulation ([18]; [45]). Recent work has emphasized fathers’ contribution to this process. In particular, paternal responses to children’s negative affect, such as offering guidance, validation, and problem-focused discussion, appear to support adolescents’ development of adaptive regulatory strategies. Adolescents who experience this form of paternal emotional availability are more likely to use cognitive reappraisal, an approach consistently associated with fewer depressive symptoms ([60]). Studies also suggest that emotionally present fathers foster a stronger sense of personal efficacy and relational security, which are linked to more flexible and constructive emotional responding ([17]). Better emotion regulation, in turn, serves as a proximal mechanism through which paternal involvement relates to adolescent mental health. For instance, secure father–child relationships have been associated with reduced depressive symptoms partially through enhanced reappraisal capacity ([30]).

Taken together, these findings highlight the use of emotion regulation strategies as a key mechanism that may help explain how father presence shapes adolescent psychological adjustment. Building on this evidence, we hypothesized that adolescents who report higher levels of father presence would demonstrate greater engagement in adaptive emotion regulation strategy (i.e., cognitive reappraisal) and less likely to rely on maladaptive strategy (i.e., expressive suppression), and that these strategies would mediate the longitudinal link between father presence and depressive symptoms.

### 1.4. Chain Mediation of Emotion Beliefs and Emotion Regulation Strategies

Emotion beliefs and emotion regulation are conceptually linked, in that beliefs about the controllability and usefulness of emotions seem to shape how adolescents approach regulation ([21]). When emotions are viewed as controllable, individuals are more inclined to engage actively with their emotional experiences rather than avoid them ([36]). They are also more likely to adopt adaptive strategies, such as cognitive reappraisal, which have been linked to lower levels of anxiety and depression ([55]).

According to [24]’s ([24]) process model of emotion regulation, regulating emotions involves multiple steps: recognizing the need to regulate, choosing suitable strategies, applying them, and evaluating their effectiveness. Beliefs about the nature and usefulness of emotions influence decisions at each stage, shaping which strategies adolescents choose and how effectively they are applied ([3]; [21]; [24]). Subsequent research has demonstrated that emotion beliefs not only determine whether individuals are willing to engage in emotion regulation, but also influence their choice of strategies and the flexibility and effectiveness of applying these strategies ([21]; [55]). For example, individuals who perceive emotions as changeable are more inclined to employ proactive strategies such as cognitive reappraisal, while those who regard emotions as uncontrollable tend to resort to the suppression of emotional expression ([47]). Empirical studies further indicate that greater use of cognitive reappraisal predicts lower depressive symptoms ([26]; [30]).

Taken together, these finding suggest that emotion beliefs provide a cognitive foundation that shapes adolescents’ regulatory choices, which may help explain why youth with more adaptive beliefs tend to report fewer depressive symptoms. Building on this evidence, the present study proposes that emotion beliefs and emotion regulation operate in sequence. Specifically, we expected that adolescents with higher perceived father presence would report more adaptive emotion beliefs, which would in turn be associated with greater use of cognitive reappraisal, ultimately predicting lower depressive symptoms. This hypothesis reflects the cognitive–emotional pathway at the center of our conceptual model.

### 1.5. The Current Study

In recent years, the role of fathers in shaping adolescents’ mental health has gained increased attention. While much of the earlier research focused primarily on the role of mothers, a growing body of work now highlights the distinct and essential role fathers play in providing emotional support and facilitating socialization during adolescence. Empirical research consistently demonstrates that higher levels of perceived paternal presence are inversely associated with depressive symptoms in adolescents ([14], [15]; [31]). However, longitudinal studies examining the longitudinal impact of paternal presence on depressive symptoms in adolescents remain relatively rare. Moreover, the mediating processes through which paternal presence exerts its influence, and the way in which these processes interact dynamically over time, have yet to be clarified.

Building on previous work, this study focuses on two mediating mechanisms that are theoretically grounded and may collectively explain the longitudinal relationship between paternal presence and depressive symptoms in adolescents, which are emotion beliefs and emotion regulation strategies. Specifically, drawing on emotional socialization theory and emotion regulation theory, the present study aims to examine whether emotion beliefs and emotion regulation strategies exert independent and sequential effects in mediating the effect of paternal presence on depressive symptoms in adolescents. The hypotheses of the current study are proposed as follows. First, father presence would negatively predict adolescents’ depressive symptoms (Hypothesis 1). Second, adolescents’ emotion beliefs and emotion regulation strategies would independently mediate the longitudinal relationship between father presence and depressive symptoms (Hypothesis 2). Finally, emotion beliefs and emotion regulation strategies would jointly operate in sequence, such that father presence would predict more adaptive emotion beliefs, which would be related to their use of emotion regulation strategies, ultimately predicting lower depressive symptoms (Hypothesis 3).

## 2. Materials and Methods

### 2.1. Participants and Procedure

This study utilized data from three waves of the Family and Child Development Project (FCDP), which involved participants from two high schools located in Shandong Province, China. Data were gathered through paper-based questionnaires administered in classrooms by trained research assistants. A total of 1074 high school students participated in the study, with data collected at three time points: June 2024 (T1), December 2024 (T2), and June 2025 (T3). A six-month interval was chosen based on prior longitudinal research showing that adolescence is a critical time for shifts in relational dynamics with parents ([13]), and adolescent emotion beliefs, regulation, and depressive symptoms also exhibit detectable changes over medium-term periods ([8]; [37]). While a longer timeline could provide additional insights, a six-month interval allowed us to effectively balance the need for longitudinal data with the challenges of participant retention and data quality in this age group ([19]). At T1, the average age of the participants was 16.06 years (*SD* = 0.43), with 47.11% identifying as male and 52.89% as female. In addition to the primary variables of interest, demographic information was also collected, including participants’ age, gender (0 = male, 1 = female, 2 = other), parental education level (1 = elementary school or below, 2 = middle school, 3 = high school or vocational training, 4 = undergraduate degree, 5 = graduate degree or higher), and family income (1 = 1000 RMB and below; 2 = 1001–3000 RMB; 3 = 3001–5000 RMB; 4 = 5000–10,000 RMB; 5 = 10,001–20,000 RMB; 6 = 20,000 RMB and above). Both adolescent participants and their parents provided written informed consent, and participants received compensation for their involvement at the conclusion of the study. The study procedures and assessments were approved by the Ethics Committee of the University, with IRB approval number IRB#202409067b.

### 2.2. Measures

#### 2.2.1. Father Presence

At T1, adolescents completed the 31-item Chinese version of the Father Presence Questionnaire-Short Form, which was originally developed by [38] ([38]), later revised by [2] ([2]), and validated for Chinese adolescents by [40] ([40]). This scale composes three subscales, including relationship with the father (e.g., “I felt my father was behind me and supported my choices or activities”), beliefs about the father (e.g., “Fathers affect their sons’ and daughters’ moral values or behavior”), and intergenerational family influences (e.g., “My father felt warm and safe when he was with his father”). Respondents rated items on a 5-point Likert scale (1 = never, 2 = seldom, 3 = occasionally, 4 = frequently, 5 = always). Higher scores on the average item ratings indicated greater perceived father presence. The scale showed excellent reliability in this study (α = 0.93).

#### 2.2.2. Emotion Beliefs

At T2, each adolescent responded to the 16-item Emotion Beliefs Questionnaire ([3]). This scale assesses two key dimensions of emotion beliefs, including beliefs about the controllability of emotions and beliefs about the usefulness of emotions. These beliefs are assessed for negative emotions (e.g., sadness) and positive emotions (e.g., happiness). Sample items include “Once people are experiencing negative emotions, there is nothing they can do about modifying them” and “There is very little use for positive emotions.” Responses were rated on a 7-point Likert scale (1 = strongly disagree, 7 = strongly agree). Higher scores on the average ratings reflect more maladaptive beliefs, indicating stronger perceptions of emotions as uncontrollable and unhelpful. The scale demonstrated high reliability in this study (α = 0.94).

#### 2.2.3. Emotion Regulation

At T2, participants completed the 10-item Emotion Regulation Questionnaire (ERQ; [25]), which measures adolescents’ use of cognitive reappraisal (e.g., “I control my emotions by changing the way I think about the situation I’m in”) and expressive suppression (e.g., “When I am feeling negative emotions, I make sure not to express them”) in their attempts to regulate emotions. All items were rated on a 7-point Likert scale (1 = strongly disagree, 7 = strongly agree). The Chinese version of this scale has previously been shown to be reliable ([27]; [59]). In the present study, the scale demonstrated a reliability of 0.92 for the cognitive reappraisal subscale, and 0.77 for the expressive suppression subscale.

#### 2.2.4. Depressive Symptoms

At T3, each adolescent responded to the short form of the Center for Epidemiologic Studies Depression Scale (CES-D; [1]). Sample items include “I had trouble keeping my mind on what I was doing” and “I felt I could not shake off the blues even with help from my family and friends.” They rated their responses using a 4-point scale (1 = rarely; 4 = most or all of the time). Scores were averaged, with higher scores indicating greater depressive symptom severity in the past week. The reliability and validity of the Chinese version of the CES-D scale have been supported in prior research ([7]). This scale demonstrated good reliability in the current study (α = 0.84).

#### 2.2.5. Covariates

Family income, parent education, as well as adolescent age and gender were used as covariates in the analyses.

### 2.3. Analysis Plan

Data analysis was conducted in lavaan package (0.6–20; [53]) using R (version 4.0.1; [51]). The sample sizes were 1074, 1058, and 985, for waves one, two, and three, respectively. The percentage of missingness on any study variable ranges from 0.00% to 23.62%, with adolescents’ depressive symptoms at T3 having the highest percentage of missingness. Little’s Missing Completely at Random (MCAR) indicated that the data were missing completely at random, χ^2^_(25)_ = 29.25, *p* = 0.252. Full Information Maximum Likelihood (FIML) was used to handle missingness in the subsequent analyses to make full use of available data, following guidelines by [46] ([46]).

Descriptive statistics were first calculated to explore the distribution of all study variables, including measures of central tendency (means), variability (standard deviations), and normality (skewness and kurtosis). Bivariate correlations were then computed to examine the relationships among the covariates and key study variables. Next, path analysis was conducted to test the proposed chain mediation model, which posits that father presence influences adolescent depressive symptoms through a sequential process involving emotion beliefs and emotion regulation strategies (specifically, reappraisal and suppression). The analysis proceeded in the following three steps. First, a direct effects model (Model A) was tested, where adolescent depressive symptoms was regressed onto father presence without including any mediators. Second, Model B added emotion beliefs as a single mediator, and Model C added emotion regulation strategies as a single mediator. This model included only those paths directly implied by Hypothesis 2. Next, to examine whether emotion beliefs and emotion regulation operate better in parallel or in sequence, we specified and compared two mediation models. Specifically, Model D (Parallel Mediation Model) allowed emotion beliefs and emotion regulation strategies to function as independent, simultaneous mediators of the association between father presence and depressive symptoms. Model E (Serial Mediation Model) specified a theory-driven sequential pathway in which emotion beliefs preceded emotion regulation strategies, reflecting the assumption that beliefs about emotions provide a cognitive foundation for subsequent regulatory strategy selection (Hypothesis 3). Note that both emotion beliefs and emotion regulation strategies were assessed at T2 in the present study. Therefore, the sequential pathway tested in the serial mediation model reflects a theory-based ordering assumption rather than an empirically verified temporal process. While the direction from emotion beliefs to regulatory strategy selection is well-supported by theoretical and empirical literature, the contemporaneous timing of measurement limits the ability to infer temporal precedence.

The significance of both the direct and indirect effects was assessed using maximum likelihood estimation with bias-corrected bootstrapping, with a 95% confidence interval calculated from 10,000 resamples. Given the sensitivity of chi-square index to sample size ([34]), model fit was evaluated based on values of comparative fit index (CFI > 0.90), root mean square error of approximation (RMSEA < 0.08), and standardized root mean square residual (SRMR < 0.08; [29]).

Finally, during the preparation of the manuscript, the authors used ChatGPT 5.1 in order to improve the language of the manuscript, such as checking for grammar issues, without it making any contribution to the content.

## 3. Results

### 3.1. Descriptive Analyses

Table 1 presents the means, standard deviations, and correlation matrix of study variables. Bivariate correlations revealed significant associations between the key variables. Specifically, father presence at T1 was negatively associated with adolescent depressive symptoms at T3 (r = −0.20, *p* < 0.001), negatively related to T2 emotion beliefs (r = −0.22, *p* < 0.001), positively related to T2 cognitive reappraisal (r = 0.15, *p* < 0.001), and negatively associated with T2 expressive suppression (r = −0.10, *p* = 0.003). T2 emotion beliefs were positively related to depressive symptoms at T3 (r = 0.39, *p* < 0.001), negatively related to T2 cognitive reappraisal (r = −0.35, *p* < 0.001), and positively associated with T2 expressive suppression (r = 0.19, *p* < 0.001). T2 cognitive reappraisal was positively related to T2 expressive suppression (r = 0.17, *p* < 0.001) and negatively related to depressive symptoms at T3 (r = −0.28, *p* < 0.001). Finally, T2 expressive suppression was positively associated with depressive symptoms at T3 (r = 0.10, *p* = 0.007). For covariates, family income, maternal and paternal education, and adolescent age and sex were associated with at least one key study variable, and were therefore retained as covariates in the path models (see Table 1 for details).

### 3.2. Primary Analyses

Table 2 and Table 3 provide the standardized estimates of all path coefficients for both the direct and indirect paths in Models A, B, C, D, and E. First, Model A examined the predictive role of father presence at T1 on depressive symptoms at T3. The results indicated that higher levels of T1 father presence negatively predicted adolescent depressive symptoms at T3 (β = −0.20, *p* < 0.001, 95% CI [−0.26, −0.13]). Models B and C then separately examined the mediating role of emotion beliefs (Model B) and emotion regulation strategies (Model C), linking father presence at T1 and depressive symptoms at T3. Both models fit the data well (RMSEA = 0.000 [0.000, 0.033], CFI = 1.000, SRMR = 0.010 for Model B; RMSEA = 0.023 [0.000, 0.044], CFI = 0.963, SRMR = 0.016 for model C). Results from Model B indicated that higher father presence at T1 predicted lower levels of maladaptive emotion beliefs at T2 (β = −0.22, *p* < 0.001, 95% CI [−0.28, −0.16]), which in turn predicted lower levels of depression (β = 0.36, *p* < 0.001, 95% CI [0.30, 0.42]). The indirect path from father presence to depressive symptoms via emotion beliefs was significant (β = −0.08, *p* < 0.001, 95% CI [−0.11, −0.05]). Model C revealed that higher father presence at T1 was related to more use of cognitive reappraisal at T2 (β = 0.15, *p* < 0.001, 95% CI [0.08, 0.21]), which then predicted fewer depressive symptoms (β = −0.30, *p* < 0.001, 95% CI [−0.35, −0.21]). The indirect path from father presence to depressive symptoms via cognitive reappraisal was significant (β = −0.04, *p* < 0.001, 95% CI [−0.06, −0.02]).

Finally, Models D (parallel mediation model) and E (serial mediation model) considered the mediating effect of both emotion beliefs and emotion regulation strategies. Both models showed identical and good fit to data, RMSEA = 0.018 [0.000, 0.036], CFI = 0.987, and SRMR = 0.017. Results from the parallel mediation model (Model D) indicated that greater father presence at T1 predicted fewer maladaptive emotion beliefs at T2 (β = −0.22, *p* < 0.001, 95% CI [−0.28, −0.16]), higher levels of cognitive reappraisal at T2 (β = 0.15, *p* < 0.001, 95% CI [0.08, 0.21]), lower levels of expressive suppression at T2 (β = −0.10, *p* = 0.003, 95% CI [−0.16, −0.03]), and fewer depressive symptoms at T3 (β = −0.09, *p* = 0.012, 95% CI [−0.15, −0.02]). Additionally, more maladaptive emotion beliefs at T2 predicted more depressive symptoms at T3 (β = 0.29, *p* < 0.001, 95% CI [0.22, 0.36]). T2 cognitive reappraisal predicted fewer depressive symptoms at T3 (β = −0.17, *p* < 0.001, 95% CI [−0.24, −0.10]). The indirect paths from father presence to depressive symptoms were significant via emotion beliefs (β = −0.07, *p* < 0.001, 95% CI [−0.09, −0.04]) and via cognitive reappraisal (β = −0.03, *p* = 0.001, 95% CI [−0.04, −0.01]). Expressive suppression did not significantly mediate the association between father presence and depressive symptoms (β = −0.01, *p* = 0.132, 95% CI [−0.01, 0.00]).

Findings from the serial mediation model (Model E; Figure 1) indicated that greater father presence at T1 predicted fewer maladaptive emotion beliefs at T2 (β = −0.22, *p* < 0.001, 95% CI [−0.28, −0.16]), higher levels of cognitive reappraisal at T2 (β = 0.07, *p* = 0.024, 95% CI [0.01, 0.13]), and fewer depressive symptoms at T3 (β = −0.09, *p* = 0.012, 95% CI [−0.15, −0.02]). Additionally, more maladaptive emotion beliefs at T2 were negatively associated with T2 cognitive reappraisal (β = −0.33, *p* < 0.001, 95% CI [−0.39, −0.27]), positively related to T2 expressive suppression (β = 0.18, *p* < 0.001, 95% CI [0.11, 0.24]), and predicted more depressive symptoms at T3 (β = 0.29, *p* < 0.001, 95% CI [0.22, 0.36]). Finally, T2 cognitive reappraisal predicted fewer depressive symptoms at T3 (β = −0.17, *p* < 0.001, 95% CI [−0.24, −0.10]). The paths from T1 father presence to T2 expressive suppression and from T2 expressive suppression to T3 depressive symptoms were not significant. The indirect paths from father presence to depressive symptoms were significant via emotion beliefs (β = −0.07, *p* < 0.001, 95% CI [−0.09, −0.04]) and via cognitive reappraisal (β = −0.01, *p* = 0.043, 95% CI [−0.02, −0.00]) independently. Additionally, father presence was associated with fewer depressive symptoms through a serial mediation, from emotion beliefs to cognitive reappraisal (β = −0.01, *p* < 0.001, 95% CI [−0.02, −0.01]). Expressive suppression did not significantly mediate the association between father presence and depressive symptoms (β = −0.00, *p* = 0.217, 95% CI [−0.01, 0.00]) nor did it contribute to the sequential pathway from father presence through emotion beliefs to depressive symptoms (β = −0.00, *p* = 0.106, 95% CI [−0.01, 0.00]). Note that because model comparison between the parallel mediation model (Model D) and the serial mediation model (Model E) indicated identical model fit, the data did not empirically favor one mediation structure over the other. We therefore retained the serial mediation model for interpretative emphasis based on theoretical considerations rather than statistical superiority.

## 4. Discussion

An increasing body of research has established that father presence is negatively correlated with adolescent depressive symptoms. However, longitudinal studies examining both the direct and indirect effects of father presence on depression remain limited. To the best of our knowledge, this study is the first to employ a longitudinal design to explore the mediating roles of emotion beliefs and emotion regulation strategies in the relationship between father presence and adolescent depressive symptoms. Our findings demonstrate that father presence influences depressive symptoms directly, as well as indirectly through emotion beliefs alone and cognitive reappraisal alone, as well as via the sequential mediation of emotion beliefs and cognitive reappraisal. These results deepen our understanding of the mechanisms linking father presence to depressive symptoms, and offer valuable theoretical insights for interventions aimed at preventing depression during adolescence.

### 4.1. The Influence of Father Presence on Depressive Symptoms

Our study confirmed Hypothesis 1, revealing that father presence was negatively associated with adolescent depressive symptoms. Specifically, adolescents who perceived their fathers as emotionally invested and psychologically available reported lower depressive symptoms over time. This longitudinal finding provides evidence for the protective role of father presence against depressive symptoms, reinforcing the conclusions drawn from previous cross-sectional research ([48]; [49]). According to the theory of father presence ([38]), paternal presence encompasses not only physical proximity but also emotional investment and psychological availability. This emotional presence provides adolescents with the necessary security, sense of belonging, and emotional support, reducing their risk of developing internalizing problems ([12]; [14]). Collectively, these findings highlight the foundational role of father presence in adolescents’ psychological development and provide a basis for future family-based interventions that aimed at enhancing paternal involvement and promoting adolescent mental health.

### 4.2. The Mediating Effect of Emotion Beliefs

The findings demonstrated that emotion beliefs served as a significant mediator in the association between father presence and adolescents’ depressive symptoms. Specifically, father presence predicted fewer maladaptive emotion beliefs in adolescents, which in turn contributed to lower subsequent levels of depressive symptoms. This model suggests that the presence of the father reduces the risk of adolescents developing depressive symptoms by promoting more adaptive emotion beliefs, namely that emotions are understandable, controllable, and have functional value. These findings are consistent with the triadic model of emotions ([45]), which emphasizes the role of parents, particularly fathers, in regulating adolescents’ emotions. Fathers influence not only the direct emotional responses of adolescents but also their beliefs about the controllability of emotions, shaping their motivation to regulate emotions effectively ([21]). Within this theoretical framework, parents’ active emotional communication, sensitivity, and responsive interactions play a crucial role in promoting adolescents’ social and emotional competencies ([21]). Through active emotional communication, sensitivity, and responsiveness, fathers help adolescents develop a belief system that views emotions as controllable and functional. Such beliefs enable adolescents to better manage negative affect, effectively cope with stress, and, ultimately, exhibit fewer depressive symptoms ([21]; [55]). Collectively, these findings provided support for Hypothesis 2, indicating that father presence indirectly alleviated adolescents’ depressive symptoms by promoting adaptive emotion beliefs.

### 4.3. The Mediating Role of Emotion Regulation

The findings revealed that father presence positively predicted adolescents’ use of cognitive reappraisal strategies, which in turn negatively predicted depressive symptoms. However, the mediating effect of expressive suppression was not significant. These results suggest that father presence may buffer adolescents against depressive symptoms particularly through enhancing their use of adaptive emotion regulation strategies, which is cognitive reappraisal. Parental involvement often manifests as emotional support and guidance, which helps adolescents develop more flexible and adaptive strategies for managing negative emotions ([49]). Research indicates that active paternal involvement enhances adolescents’ ability to utilize cognitive reappraisal to process negative emotional states ([45]; [60]). Furthermore, the unique function of father–child interactions, which often involves motivation and challenge, provides opportunities for adolescents to practice problem-solving, derive meaning from emotional experiences, and regulate their emotions ([52]). These experiential interactions directly contribute to the development of cognitive reappraisal, which ultimately reduce the risk of depressive symptoms.

### 4.4. The Sequential Mediating Role of Emotion Beliefs and Emotion Regulation

Our results were consistent with Hypothesis 3, indicating that emotion beliefs and cognitive reappraisal can be organized as a theoretically meaningful sequential pathway linking father presence to adolescent depressive symptoms. Specifically, father presence reduced adolescents’ beliefs that emotions are uncontrollable and useless, which in turn relate to their use of cognitive reappraisal, ultimately decreasing depressive symptoms. Importantly, although both parallel and serial mediation models demonstrated equivalent statistical fit, the serial specification offers a more integrative theoretical account by positioning emotion beliefs as a cognitive antecedent that shapes adolescents’ selection of emotion regulation strategies. Theoretically, this finding integrates the emotion socialization framework ([18]) with the emotion regulation process model ([24]), proposing that fathers influence adolescent mental health by altering core emotion beliefs. Through modeling, responsiveness, and communication, fathers help adolescents develop the ability to see emotions as manageable, which is associated with more frequent employment of cognitive reappraisal in stressful situations ([21]). This process reduces the accumulation of negative affect and mitigates the risk of depression. Meta-analytic evidence further supports this idea, showing that cognitive reappraisal is associated with better psychological well-being and lower levels of depressive and anxiety symptoms ([61]). This sequential pathway reveals the underlying cognitive and emotional mechanisms through which father presence influences depressive symptoms, offering a new perspective on the unique role of fathers in adolescent emotional socialization.

Notably, the two emotion regulation strategies examined in the present study yielded different patterns of association. Whereas cognitive reappraisal functioned as both an independent and sequential mediator, expressive suppression did not emerge as a significant mediating pathway linking father presence to depressive symptoms. This divergence underscores the importance of distinguishing between qualitatively different regulation strategies when examining family-based emotional socialization processes ([24]; [21]). From an emotion socialization perspective, paternal emotional presence may be more strongly linked to adolescents’ cognitive understanding and reinterpretation of emotional experiences, rather than to the inhibition of emotional expression. Fathers who are emotionally available may encourage adolescents to reflect on, reframe, and make sense of emotional experiences, which aligns more closely with cognitive reappraisal than with suppression ([18]; [45]; [21]). In contrast, expressive suppression is often shaped by broader cultural norms, peer contexts, or situational demands, and may therefore be less sensitive to variations in paternal emotional involvement.

In addition, the nonsignificant role of expressive suppression may be understood within the cultural context. In East Asian societies, including China, emotional restraint and suppression are often socially normative and encouraged to maintain interpersonal harmony ([57]). For Chinese adolescents, suppressing emotions may be seen as more socially acceptable than expressing or disclosing emotional struggles ([16]). Thus, the relationship between emotional expression suppression and depressive symptoms may be more context-dependent and vary across cultures ([57]; [5]). This finding highlights the importance of considering cultural nuances when evaluating the effectiveness of emotion regulation strategies.

It should be noted that although emotion beliefs and emotion regulation strategies were modeled in a sequential order, these variables were assessed concurrently at T2. Thus, the proposed sequence reflects a theoretically grounded ordering rather than an empirically established temporal process. From a developmental perspective, these findings highlight a theoretically coherent pattern of associations in which adolescents’ beliefs about emotions are closely linked to their use of emotion regulation strategies, ultimately influencing emotional outcomes. These results provide support for Hypothesis 3, offering a valuable cognitive–emotional model for understanding family-based emotional socialization processes. Moreover, this model provides important implications for designing family interventions aimed at improving father involvement and promoting adolescent mental health.

### 4.5. Implications and Limitations

The findings of this study have meaningful theoretical and applied implications. In many Asian families, fathers often assume the role of financial provider, whereas mothers are commonly responsible for daily caregiving and emotional communication ([39]). This division of roles may contribute to reduced paternal involvement in emotional socialization, limiting adolescents’ opportunities to receive emotional support or guidance from their fathers. Prior longitudinal research has shown that father absence during childhood is associated with elevated trajectories of depressive symptoms throughout adolescence and into adulthood ([15]). Building on this work, the present study demonstrates that even within sociocultural contexts where paternal involvement is relatively constrained, adolescents’ subjective perceptions of their fathers’ emotional availability and psychological presence meaningfully predict their mental health through cognitive and emotional pathways. These findings suggest that family-based interventions should extend beyond merely increasing paternal physical presence. Instead, enhancing fathers’ emotional responsiveness and psychological accessibility may have greater impact on adolescents’ emotional development and well-being. Strengthening adolescents’ perception of paternal emotional support may foster healthier emotion socialization processes, ultimately reducing depressive risk.

Despite its contributions, this study has several limitations. First, although a three-wave longitudinal design was used, the dynamic interplay among variables was not examined in depth. Future research may adopt cross-lagged panel modeling or latent growth modeling to more rigorously examine causal directions and developmental trajectories. Second, all measures relied on adolescent self-report, which may introduce response bias. Future studies may incorporate multiple informants, behavioral observations, or physiological measures to improve methodological rigor. Third, given the large sample size, statistical significance should be interpreted alongside standardized effect sizes and confidence intervals. Small effects at the individual level may nevertheless accumulate over time and have meaningful implications at the population or developmental level. Fourth, because emotion beliefs and emotion regulation strategies were measured in the same wave, the sequential mediation pathway cannot be interpreted as reflecting a strictly time-ordered process. Future longitudinal work should incorporate cross-wave measurement of mediators to more directly establish temporal directionality. Finally, because the sample consisted solely of Chinese adolescents, the generalizability of the findings should be tested in other cultural contexts. Given evidence that paternal involvement during early childhood may exert long-term influence on developmental outcomes, future research may also explore whether the implications of father presence differ at various developmental stages. Cross-cultural comparative research would help distinguish universal patterns of paternal influence from those driven by specific cultural values or parenting norms.

## 5. Conclusions

Using a three-wave longitudinal design, this study examined how father presence relates to adolescent depressive symptoms over time, and whether emotion beliefs and emotion regulation mediate this association. The findings indicate that both emotion beliefs and cognitive reappraisal operate as independent mediators and as sequential mediators linking father presence to lower depressive symptoms. Specifically, greater father presence predicted fewer maladaptive beliefs about emotions being uncontrollable or useless, which in turn supported adolescents’ use of cognitive reappraisal, ultimately reducing depressive symptoms.

These findings extend the theoretical foundation of father presence by identifying cognitive and emotional mechanisms through which paternal involvement affects adolescent mental health. From a practical perspective, the results highlight the importance of strengthening fathers’ emotional engagement within family-based mental health interventions. By fostering adaptive emotion beliefs and effective emotion regulation strategies, fathers may play a meaningful role in supporting adolescents’ psychological well-being. Future research should continue to test this mediation model across diverse cultural and developmental contexts to better understand how cultural norms, timing of paternal involvement, and broader ecological influences shape these processes. Such work will enhance our understanding of adolescent emotional development and inform more culturally responsive and developmentally attuned intervention approaches.

## Figures and Tables

**Figure 1 behavsci-16-00047-f001:**
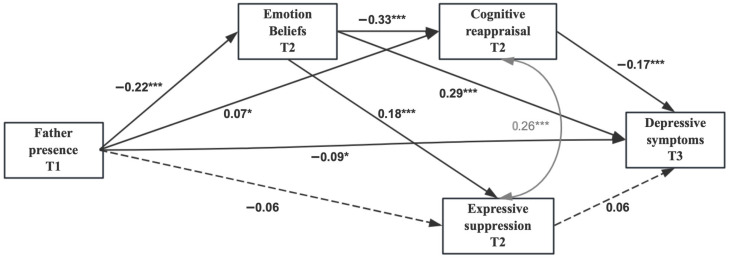
Path Model of the Relations between Father Presence, Emotion beliefs, Cognitive Reappraisal, Expressive Suppression, and Depressive Symptoms (Model E). Note. Standardized coefficients were presented. The paths of covariates were hidden for visual clarity. The dotted line indicates that the path coefficient is nonsignificant. T1 = wave one, T2 = wave two, T3 = wave three. * *p* < 0.05. *** *p* < 0.001.

**Table 1 behavsci-16-00047-t001:** Means, Standard Deviations, and Correlations of Study Variables.

Variables	*M*	*SD*	1	2	3	4	5	6	7	8	9	10
Key variables												
1. Father presence T1	3.52	0.67	–									
2. Emotion beliefs T2	2.26	1.04	−0.23 **	–								
3. Cognitive reappraisal T2	4.96	1.28	0.15 **	−0.35 **	–							
4. Expressive suppression T2	3.64	1.30	−0.10 **	0.19 **	0.17 **	–						
5. Depressive symptoms T3	1.95	0.54	−0.20 **	0.39 **	−0.28 **	0.10 **	–					
Covariates												
6. Family income	4.03	1.09	0.11 **	−0.01	0.04	−0.04	−0.08 *	–				
7. Mom education	2.87	0.93	0.10 **	0.004	0.05	−0.05	−0.08 *	0.27 **	–			
8. Dad education	2.97	0.89	0.15 **	0.01	0.03	0.01	−0.08 *	0.26 **	0.61 **	–		
9. Age	16.06	0.43	−0.01	0.02	−0.02	−0.04	0.03	−0.07 *	−0.02	−0.06	–	
10. Gender			−0.14	−0.01	−0.01	−0.07 *	−0.05	0.01	0.06	0.04	−0.05	–

Note. T1 = wave one, T2 = wave two, T3 = wave three. * *p* < 0.05. ** *p* < 0.01.

**Table 2 behavsci-16-00047-t002:** Standardized Bootstrap Estimates and 95% Bias-corrected CI for Direct and Indirect Effects in Path Models A, B, and C.

	Model A			Model B			Model C		
		95% CI			95% CI			95% CI	
Effect	β	LL	UL	β	LL	UL	β	LL	UL
Direct									
Father presence T1 → Depressive symptoms T3	−0.20 ***	−0.26	−0.13	−0.10 **	−0.17	−0.03	−0.14 ***	−0.21	−0.07
Indirect									
Father presence T1 → Emotion beliefs T2 → Depressive symptoms T3				−0.08 ***	−0.11	−0.05			
Father presence T1 → Cognitive reappraisal T2 → Depressive symptoms T3							−0.04 ***	−0.06	−0.02
Father presence T1 → Expressive suppression T2 → Depressive symptoms T3							−0.01	−0.02	0.00

Note. Standardized parameter estimates and 95% confidence intervals were presented. CI = confidence interval, LL = lower limit, UL = upper limit, T1 = wave one, T2 = wave two, T3 = wave three. ** *p* < 0.01. *** *p* < 0.001.

**Table 3 behavsci-16-00047-t003:** Standardized Bootstrap Estimates and 95% Bias-corrected CI for Direct and Indirect Effects in Path Models D and E.

	Model D (Parallel Model)			Model E (Serial Model)		
		95% CI			95% CI	
Effect	β	LL	UL	β	LL	UL
Direct						
Father presence T1 → Depressive symptoms T3	−0.09 *	−0.15	−0.02	−0.09 *	−0.15	−0.02
Indirect						
Father presence T1 → Emotion beliefs T2 → Depressive symptoms T3	−0.07 ***	−0.09	−0.04	−0.07 ***	−0.09	−0.04
Father presence T1 → Cognitive reappraisal T2 → Depressive symptoms T3	−0.03 **	−0.04	−0.01	−0.01 *	−0.02	0.00
Father presence T1 → Expressive suppression T2 → Depressive symptoms T3	−0.01	−0.01	0.00	−0.00	−0.01	0.00
Father presence T1 → Emotion beliefs T2 → Cognitive reappraisal T2 → Depressive symptoms T3				−0.01 ***	−0.02	−0.01
Father presence T1 → Emotion beliefs T2 → Expressive suppression T2 → Depressive symptoms T3				−0.00	−0.01	0.00

Note. Standardized parameter estimates and 95% confidence intervals were presented. CI = confidence interval, LL = lower limit, UL = upper limit, T1 = wave one, T2 = wave two, T3 = wave three. * *p* < 0.05. ** *p* < 0.01. *** *p* < 0.001.

## Data Availability

The data that support the findings of this study are available from the corresponding author upon reasonable request.

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
