# Peer review of "The Longitudinal Impact of Father Presence on Adolescent Depressive Symptoms: The Mediating Role of Emotion Beliefs and Emotion Regulation"

_behavsci, 2025, doi:10.3390/bs16010047_

Round 1
Reviewer 1 Report
Comments and Suggestions for Authors
This is a well-written and competently analyzed paper. My only observations related to the potential impact of attrition bias over time and issues of effect size/statistical power given the large sample size-path coefficients can be statistically different from zero yet be small in magnitude and consequently less meaningful. The authors need to address these 2 ares of concern.
Author Response
behavsci-4019117
The Longitudinal Impact of Father Presence on Adolescent Depressive Symptoms: The Mediating Role of Emotion Beliefs and Emotion Regulation
First and foremost, we would like to express our gratitude to the editor and reviewers for allowing us to revise our manuscript. Your insightful comments have highlighted important areas for refinement and have significantly contributed to enhancing the quality and rigor of our work. Below, we address each of your comments in detail. We hope that the revised version addresses all concerns satisfactorily and we are happy to provide further clarification if needed.
Reviewer 1’s Comments
This is a well-written and competently analyzed paper. My only observations related to the potential impact of attrition bias over time and issues of effect size/statistical power given the large sample size-path coefficients can be statistically different from zero yet be small in magnitude and consequently less meaningful. The authors need to address these 2 ares of concern.
Comment 1: My only observations related to the potential impact of attrition bias over time.
Response: Thank you for your thoughtful comment regarding potential attrition bias. In the revised manuscript, we now report wave-specific sample sizes (T1 = 1,074; T2 = 1,058; T3 = 985) and explicitly describe our missing-data handling strategy (please see the first paragraph in section 2.3 Analysis plan, page 7). Path analyses were conducted in lavaan using Full Information Maximum Likelihood (FIML) to make full use of available data, following guidelines by Newman (2014). In addition, we conducted Little’s Missing Completely at Random (MCAR; Little, 1988), and found that data were missing completely at random, χ2(25) = 29.25, p = .25. These details have now been added to the Analysis plan (page 7, lines 311-318).
Comment 2: The issues of effect size/statistical power given the large sample size-path coefficients can be statistically different from zero yet be small in magnitude and consequently less meaningful.
Response: Thank you for highlighting this important point. We agree with this observation. Accordingly, we have revised the manuscript to place greater emphasis on standardized effect sizes and confidence intervals rather than statistical significance alone. We also clarify that the observed effects, although modest in magnitude, represent developmentally meaningful associations consistent with prior longitudinal studies of parental involvement and adolescent mental health. This clarification has been incorporated into the Discussion section. For example, we have added “given the large sample size, statistical significance should be interpreted alongside standardized effect sizes and confidence intervals. Small effects at the individual level may nevertheless accumulate over time and have meaningful implications at the population or developmental level” in the Implications and Limitations section (please see page 13, lines 524-528).
Reviewer 2 Report
Comments and Suggestions for Authors
I consider the topic of this study—fathers’ emotional involvement/presence and adolescents’ depressive symptoms—highly timely and potentially impactful. However, I identified multiple errors and inconsistencies in the research design, measurement description, and reporting of results. In its current form, these issues substantially undermine confidence in the manuscript, the data, and the analyses. I encourage the authors to address the points below to strengthen the study.
Comments
- Overextended framing of the core pathway in the Introduction
The logic for positioning emotional beliefs and emotion regulation as the core pathway is defensible; however, the current narrative becomes citation- and exposition-heavy, which dilutes the central proposition (perceived father emotional presence - beliefs about emotions - greater use of cognitive reappraisal - reduced depressive symptoms). The Introduction would benefit from a concise sentence or short paragraph that explicitly foregrounds this pathway and clearly articulates what the study tests.
- Weak theory-driven model specification
The authors propose a “serial (sequential) mediation” model via emotional beliefs and emotion regulation. Yet the final model figure (p. 9) reads less like a confirmatory model that tests the minimum theoretically necessary paths and more like an omnibus specification that leaves many plausible paths open. Specifically, father presence (T1) is modeled as having concurrent direct paths to emotional beliefs (T2), reappraisal (T2), and depressive symptoms (T3), and emotional beliefs (T2) also has multiple outgoing paths to reappraisal (T2), suppression (T2), and depressive symptoms (T3). In addition, Table 2 simultaneously reports multiple indirect pathways (e.g., mediation via emotional beliefs only, via reappraisal only, and via the serial pathway), which increases the risk that the analysis will be read as “drawing nearly all possible paths and then interpreting whichever ones are significant,” raising concerns about overfitting and selective interpretation.
Accordingly, the authors should (a) present a reduced model that includes only the paths that are strictly implied by the core theoretical premises, and (b) specify and compare competing models (e.g., parallel mediation vs. serial mediation) to justify why the proposed structure is the best representation of the theory. If the serial pathway is retained, the fact that the mediators are measured at the same wave (T2) weakens the empirical basis for a temporal “process.” The authors should therefore either explicitly frame the serial pathway as a theory-driven ordering assumption, or defend the ordering via time separation and/or systematic comparisons with alternative causal orderings.
- Inconsistent description of the longitudinal design (Abstract vs. Methods)
The Abstract describes the study as a “two-wave longitudinal design,” whereas the Methods clearly specify three waves (June 2024 [T1], December 2024 [T2], and June 2025 [T3]). The wave structure must be harmonized across the Abstract, Methods, and all table/figure notes. This type of discrepancy negatively affects the perceived credibility of both the manuscript and the underlying dataset.
- Conflict in reported confidence intervals for the key direct effect
In the text, Model B reports that the direct effect of father presence at T1 on depressive symptoms at T3 is significant with a CI of [-.15, -.02]. However, Table 2 (Model B) reports the CI for the same path as [-.044, .037], which includes zero. This discrepancy is serious because it reverses the conclusion (significant vs. non-significant). The authors should re-check (a) whether coefficients are standardized or unstandardized, (b) the bootstrap settings (10,000 draws; whether bias-corrected CIs were used), and (c) output/rounding conventions, and then align the numbers across the text, tables, and figures so that a single, internally consistent set of results is presented.
- Insufficient temporal justification for “serial mediation” given T2 concurrent measurement
Emotional beliefs are measured at T2, and emotion regulation (ERQ) is also measured at T2. Despite this, the manuscript strongly adopts a chain/serial mediation narrative. When mediators are measured contemporaneously within the same wave, it is difficult to claim that the study has empirically established a temporal causal sequence in which emotional beliefs precede reappraisal. The authors should either (a) restrict their interpretation to a statistical chain of associations rather than a time-ordered process, (b) present alternative model specifications (e.g., reappraisal → emotional beliefs) and compare competing models’ fit/indirect effects, or (c) if genuine sequentiality is the goal, adopt a design (or additional analyses) that separates the mediators across waves (e.g., beliefs at T2 and regulation at T3) in future work.
- Clear error in the description of the ERQ measure
The example item provided for the ERQ (emotion regulation) is “Fathers affect their sons’ and daughters’ moral values or behavior,” which is identical to an example item presented earlier for the father presence scale. The authors should thoroughly verify the Chinese version of the ERQ used, provide appropriate example items for each ERQ subscale (reappraisal/suppression), confirm scoring direction and sources, and ensure that the variables entered into the analysis are indeed the ERQ measures (including correct coding and labeling). This issue is not merely stylistic; it can be interpreted as a potential threat to measurement validity and data integrity.
- Insufficient reporting of missingness/attrition handling and estimation strategy in a multiwave panel
The manuscript states that path analyses were conducted in lavaan using ML with bias-corrected bootstrapping (10,000 draws). However, in a three-wave panel, attrition and missing data are structurally expected, yet the manuscript does not report (a) wave-specific analytic sample sizes, (b) evidence regarding selective attrition, or (c) the missing-data approach (e.g., FIML, multiple imputation, listwise deletion). Without this information, the analyses are difficult to evaluate and reproduce. The authors should report wave-specific Ns, missingness rates, and the missing-data handling strategy (in text and/or a table) to establish transparency and reproducibility.
Reviewer 3 Report
Comments and Suggestions for Authors
A central strength of the article lies in its conceptual innovation. While previous literature has documented the protective association between father involvement and adolescent mental health, the authors extend this work by proposing that emotion beliefs, youths’ perceptions of the controllability and usefulness of emotions, constitute a core cognitive mechanism linking paternal presence to psychological adjustment. This theoretical integration is relatively novel, as prior studies mostly focus on parenting behaviors, attachment, or emotional climate. I believe the theoretical section would benefit from a clearer conceptualization of what “father presence” actually entails and how it can be defined. Although the construct is introduced in the manuscript, its boundaries remain somewhat broad, combining relational experiences, internalized beliefs about the father, and intergenerational family influences. A clearer articulation of how father presence is understood in this specific research context would also help readers better interpret the proposed mechanisms linking paternal involvement to emotion beliefs and depressive symptoms.The study uses a three-wave design with data collected every six months (June 2024, December 2024, and June 2025), but it is not entirely clear why this particular interval was chosen. Offering a brief explanation for the six-month spacing, such as whether this window is thought to allow noticeable changes in emotion beliefs or regulatio, would make the methodological choices easier to follow. Providing this context would also help readers understand what kinds of developmental shifts the authors expect to observe over time and why this timeframe is considered suitable for capturing the mechanisms linking father presence to later depressive symptoms.
Author Response
behavsci-4019117
The Longitudinal Impact of Father Presence on Adolescent Depressive Symptoms: The Mediating Role of Emotion Beliefs and Emotion Regulation
First and foremost, we would like to express our gratitude to the editor and reviewers for allowing us to revise our manuscript. Your insightful comments have highlighted important areas for refinement and have significantly contributed to enhancing the quality and rigor of our work. Below, we address each of your comments in detail. We hope that the revised version addresses all concerns satisfactorily and we are happy to provide further clarification if needed.
Reviewer 3’s Comments
A central strength of the article lies in its conceptual innovation. While previous literature has documented the protective association between father involvement and adolescent mental health, the authors extend this work by proposing that emotion beliefs, youths’ perceptions of the controllability and usefulness of emotions, constitute a core cognitive mechanism linking paternal presence to psychological adjustment. This theoretical integration is relatively novel, as prior studies mostly focus on parenting behaviors, attachment, or emotional climate.
Comment 1: I believe the theoretical section would benefit from a clearer conceptualization of what “father presence” actually entails and how it can be defined. Although the construct is introduced in the manuscript, its boundaries remain somewhat broad, combining relational experiences, internalized beliefs about the father, and intergenerational family influences. A clearer articulation of how father presence is understood in this specific research context would also help readers better interpret the proposed mechanisms linking paternal involvement to emotion beliefs and depressive symptoms.
Response: Thank you for this valuable conceptual suggestion. In response, we have clarified the conceptualization of father presence and explicitly defined the construct in the manuscript. Specifically, we have clarified that father presence is defined as adolescents’ subjective perception of their fathers’ emotional availability and psychological involvement, rather than objective behavioral frequency or physical co-residence. We also more explicitly link this definition to the proposed cognitive mechanisms involving emotion beliefs and regulation (pages 2 & 3). For instance, we stated that “This conceptualization emphasizes the internal psychological experience of being emotionally supported by one’s father, which is particularly relevant for adolescents’ cognitive and emotional processing.”
Comment 2: The study uses a three-wave design with data collected every six months (June 2024, December 2024, and June 2025), but it is not entirely clear why this particular interval was chosen. Offering a brief explanation for the six-month spacing, such as whether this window is thought to allow noticeable changes in emotion beliefs or regulation, would make the methodological choices easier to follow. Providing this context would also help readers understand what kinds of developmental shifts the authors expect to observe over time and why this timeframe is considered suitable for capturing the mechanisms linking father presence to later depressive symptoms.
Response: Thank you for the insightful comment. The choice of a six-month research period is based on both theoretical and practical considerations. For theoretical consideration, research indicates that adolescence is a critical time for changes in relational dynamics with parents (Becht et al., 2017; Crocetti et al., 2017), as well as adolescents’ emotion beliefs, regulation, and depressive symptoms (Chan et al., 2023; Kökönyei et al., 2024). During this period, the individuation process often leads to fluctuations in perceived relationship quality with parents, as adolescents strive for emotional self-sufficiency (Keizer et al., 2019). Previous longitudinal work, such as a three-wave design focusing on early adolescence, has demonstrated that within relatively short time frames of six months to one year, adolescents’ perceived emotion regulation and emotional distress may exhibit dynamic changes, which suggest that emotional processes during adolescence are not static but may change rapidly within a relatively short period (Demkowicz et al., 2024). While a longer timeline could provide additional insights, a six-month interval allowed us to effectively balance the need for longitudinal data with the challenges of participant retention and data quality in this age group (Faden et al., 2004). This interval provides sufficient time to capture developmental shifts while minimizing measurement redundancy, consistent with multi-wave studies of adolescent emotional and cognitive processes. To support this argument, we have now included a discussion of these considerations on page 6, lines 247-253 of the revised manuscript. We hope this addresses your concerns and clarifies our rationale for the chosen timeline.